# SEMANTIC PRIOR FOR WEAKLY SUPERVISED CLASS-INCREMENTAL SEGMENTATION

## ABSTRACT

Class-incremental semantic image segmentation assumes multiple model updates, each enriching the model to segment new categories. This is typically carried out by providing pixel-level manual annotations for all new objects, limiting the adoption of such methods. Approaches which solely require image-level labels offer an attractive alternative, yet, such annotations lack crucial information about the location and boundary of new objects. In this paper we argue that, since classes represent not just indices but semantic entities, the conceptual relationships between them can provide valuable information that should be leveraged. We propose a weakly supervised approach that leverages such semantic relations in order to transfer some cues from the previously learned classes into the new ones, complementing the supervisory signal from image-level labels. We validate our approach on a number of continual learning tasks, and show how even a simple pairwise interaction between classes can significantly improve the segmentation mask quality of both old and new classes. We show these conclusions still hold for longer and, hence, more realistic sequences of tasks and for a challenging few-shot scenario.

## 1 INTRODUCTION

When working towards the real-world deployment of artificial intelligence systems, two main challenges arise: such systems should possess the ability to continuously learn, and this learning process should only require limited human intervention. While deep learning models have proved effective in tackling tasks for which large amounts of curated data as well as abundant computational resources are available, they still struggle to learn over continuous and potentially heterogeneous sequences of tasks, especially if supervision is limited.

In this work, we focus on the task of semantic image segmentation (SIS). A reliable and versatile SIS model should be able to seamlessly add new categories to its repertoire without forgetting about the old ones. Considering for instance a house robot or a self-driving vehicle with such segmentation capability, we would like it to be able to handle new classes without having to retrain the segmentation model from scratch. Such ability is at the core of continual learning research, the main challenge being to mitigate catastrophic forgetting of what has been previously learned (Parisi et al., 2019).

Most learning algorithms for SIS assume training samples with accurate pixel-level annotations, a time-consuming and tedious operation. We argue that this is cumbersome and severely hinders continual learning; adding new classes over time should be a lighter-weight process. This is why, here, we focus on the case where only image-level labels are provided (*e.g.*, adding the 'sheep' class comes as easily as only providing images guaranteed to contain at least a sheep). This weakly supervised task is an extremely challenging problem in itself and very few attempts have been made in the context of continual learning (Cermelli et al., 2022).

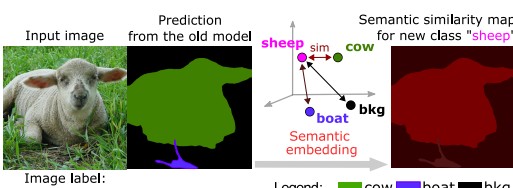

Figure 1: Our proposed Relation-aware Prior Loss (RaSP) is based on the intuition that predictions from existing classes provide valuable cues to better segment new, semantically related classes. This allows reducing supervision to image-level labels for incremental SiS.

Additionally, we argue that tackling a set of SIS tasks *incrementally* can bring opportunities to learn to segment new categories more efficiently (allowing to move from pixel-level to image-level labels) and more effectively. This can be enabled by taking into account the semantic relationship between old and new classes – as humans do. In this paper, we formalize and empirically validate a semantic prior loss that fosters such *forward transfer*. This leads to a continual learning procedure for weakly supervised SIS models that leverages the semantic similarity between class names and builds on top of the model's previous knowledge accordingly. If the model needs to additionally learn about the 'sheep' class for instance, our loss can leverage the model's effectiveness in dealing with other similar animals, such as cows, and does not need to learn it from scratch (see Fig. 1).

We validate our approach by showing that it seamlessly extends state-of-the-art SIS methods. For all our experiments, we build on the WILSON approach of Cermelli et al. (2021), buildt itself on standard techniques for weakly supervised SIS (Borenstein & Ullman, 2004; Araslanov & Roth, 2020). We extend it with our Relation-aware Semantic Prior (RaSP) loss to encourage forward transfer across classes within the learning sequence. It is designed as a simple-to-adopt regularizer that measures the similarity across old and new categories and reuses knowledge accordingly.

To summarize, our contribution is threefold. First, we propose RaSP, a semantic prior loss that treats class labels, not as mere indices, but as semantic entities whose relationship between each other matters. Second, we broaden benchmarks previously used for weakly supervised class-incremental SIS to consider both longer sequences of tasks (prior art is limited to 2, we extend to up to 11), and a few-shot setting, both with image-level annotations only. Finally, we empirically validate that the steady improvement brought by RaSP is also visible in an extended version of our approach that uses an episodic memory, filled with either past samples or web-crawled images for the old classes. We show that, in this context, the memory does not only mitigate catastrophic forgetting, but also and most importantly fosters the learning of new categories.

## 2 RELATED WORK

This work lies at the intersection of weakly supervised and class-incremental learning of SIS models. Due to the nature of our semantic prior loss, it also relates to text-guided computer vision.

**Weakly supervised SIS.** This term (Borenstein & Ullman, 2004) encompasses several tasks for which SIS models are trained using weaker annotations than the typical pixel-level labels, such as image captions, bounding boxes or scribbles. Methods assuming bounding box annotations for all relevant objects, (or produced by a pretrained detector) focus on segmenting instances within those bounding boxes (Dai et al., 2015; Ji & Veksler, 2021; Song et al., 2019; Kulharia et al., 2020). More related to our work are the methods that with image-level labels, exploiting classification activation maps (CAM) as pixel-level supervision for SIS (Zhou et al., 2016b; Kolesnikov & Lampert, 2016; Roy & Todorovic, 2017; Huang et al., 2018; Ahn & Kwak, 2018; Araslanov & Roth, 2020).

**Class-incremental SIS.** Under the hood of continual learning (Parisi et al., 2019), class-incremental learning consists in exposing a model to sequences of tasks, in which the goal is learning new classes. While most class-incremental learning methods have focused on image classification (Masana et al., 2020), some recent works started focusing on SiS (Cermelli et al., 2020; Michieli & Zanuttigh, 2021a; Douillard et al., 2021; Maracani et al., 2021; Cha et al., 2021). Yet, all aforementioned methods assume pixel-level annotations for all the new classes, which requires a huge, often prohibitively expensive amount of human work. Therefore, weakly-supervised class-incremental SIS has emerged as a viable alternative in the pioneering work of Cermelli et al. (2022), which formalizes the task, and proposes the WILSON method to tackle it. WILSON builds on top of standard weakly supervised SIS techniques, and explicitly tries to mitigate forgetting using feature-level knowledge distillation, akin to the pseudo-labeling approach of PLOP (Douillard et al., 2021).

**Text-guided computer vision.** Vision and language have a long history of benefiting from each other, and language, a modality that is inherently more semantic, has often been used as a source of supervision to guide computer vision tasks, such as learning visual representations (Quattoni et al., 2007; Gomez et al., 2017; Sariyildiz et al., 2020; Radford et al., 2021) object detection (Shi et al., 2017), zero-shot segmentation (Zhou et al., 2016b; Bucher et al., 2019; Xian et al., 2019; Li et al., 2020; Baek et al., 2021), language-driven segmentation (Zhao et al., 2017; Li et al., 2022; Ghiasi et al., 2022; Xu et al., 2022) or referring image segmentation (Hu et al., 2016; Liu et al., 2017;

Ding et al., 2021; Wang et al., 2022), among others. In some cases, this guiding process requires to create a textual embedding whose associated metrics plays a crucial role. Word2Vec (Mikolov et al., 2013), GloVe (Pennington et al., 2014) and BERT (Devlin et al., 2019) are among the most common. Similarly, our prior semantic loss assumes the availability of such similarity metrics between textual pairs. Others, such as Ghiasi et al. (2022); Xu et al. (2022); Wang et al. (2022) rely on large pre-trained text-image embeddings such as CLIP (Radford et al., 2021) or ALIGN (Jia et al., 2021) and combine those with pixel-level semantic labels to generalize SIS models for unseen concepts. Yet, none of these approaches has considered a weakly supervised incremental setting.

## 3 PROPOSED FRAMEWORK

We develop a method for Weakly Supervised Class-Incremental SIS (WSCIS). The goal is incrementally learning to segment instances from new classes by using image-level labels only, avoiding the need for pixel-level annotations. Before detailing our method, we formalize our setting.

**Problem setup and notations.** Let $\mathcal{D}^{\mathrm{b}} = \{(\mathbf{x}_k^{\mathrm{b}}, \mathbf{y}_k^{\mathrm{b}})\}_{k=1}^{N^{\mathrm{b}}}$ be a dataset for SIS, where $\mathbf{x}^{\mathrm{b}} \in \mathbb{R}^{H \times W \times 3}$ represents an input image and $\mathbf{y}^{\mathrm{b}}$ is a tensor containing the $|\mathcal{C}^{\mathrm{b}}|$-dimensional one-hot label vectors for each pixel, in a $H \times W$ spatial grid, corresponding to a set of $\mathcal{C}^{\mathrm{b}}$ semantic classes. As typical in SIS, the objects that do not belong to any of the foreground classes are annotated as a special background class ('$bkg$') – included in $\mathcal{C}^{\mathrm{b}}$. We refer to $\mathcal{D}^{\mathrm{b}}$ as the *base* task and do not make assumptions on its cardinality. $\mathcal{D}^{\mathrm{b}}$ is used to train a base model, generally defined by an encoder $E^{\mathrm{b}}$ and a decoder $F^{\mathrm{b}}$, $(E^{\mathrm{b}} \circ F^{\mathrm{b}}) \colon \mathbf{x} \to \mathbb{R}^{\mathcal{I} \times |\mathcal{C}^{\mathrm{b}}|}$, where $\mathcal{I} = H' \times W'$ is a spatial grid – corresponding to the input image size or some resized version of it – and $\mathbf{p} = (E^{\mathrm{b}} \circ F^{\mathrm{b}})(\mathbf{x})$ is the set of class prediction maps, where $p_i^c$ is the probability for the spatial location $i \in \mathcal{I}$ in the input image $\mathbf{x}$ to belong to the class $c$.

After this base training, we assume the model undergoes a sequence of learning steps, as training sets for new tasks become available. Specifically, at each learning step $t$, the model is exposed to a new set $\mathcal{D}^t = \{(\mathbf{x}_k^t, \mathbf{l}_k^t)\}_{k=1}^{N^t}$ containing $N^t$ instances labeled for previously unseen $\mathcal{C}^t$ classes, where $\mathbf{l}^t \in \mathbb{R}^{|\mathcal{C}^t|}$ is the vectorized *image-level* label corresponding to an image $\mathbf{x}^t$. Note that in each incremental step, only weak annotations (image-level labels) are provided for the new classes. This is in sharp contrast with the base task, in which the model is trained with pixel-level annotations.

The goal of WSCIS is to update the segmentation model at each incremental step $t$ in a weakly supervised way, without *forgetting* any of the previously learned classes. We learn the function $(E^t \circ F^t) \colon \mathbf{x} \to \mathbb{R}^{\mathcal{I} \times |\mathcal{Y}^t|}$, where $\mathcal{Y}^t = \bigcup_{k=1}^t \{\mathcal{C}^k\} \cup \mathcal{C}^{\mathrm{b}}$ is the set of labels at step $t$ (old and new ones). Note that, in general, we assume that data from previous tasks cannot be stored – that is, there is no episodic memory. We revisit this assumption in Sec. 4.2.

### 3.1 THE RELATION-AWARE SEMANTIC PRIOR LOSS

In this paper, we propose to leverage the semantic relationship between the new and old classes. We argue that semantic object categories are not independent, *i.e.*, the new classes $\mathcal{C}^t$ that are being learned at step $t$ may bear semantic resemblance with the old classes from $\mathcal{Y}^{t-1}$, seen by the model during previous training steps. For example, the network may have been trained to segment instances of the 'cow' class with dense supervision during the base training, and at any arbitrary incremental step $t$ the segmentation network can be tasked with learning to segment the 'sheep' class from weak-supervision. Since cow and sheep are closely related species sharing similar attributes (such as being four-legged, furry mammals), the old snapshot of the model $E^{t-1} \circ F^{t-1}$ (or, for brevity, $(E \circ F)^{t-1}$) can provide valuable cues to localize the 'sheep' regions in an image labeled as sheep, despite having never seen this animal before (see Fig. 1). Guided by this insight, instead of using the old model predictions to only give cues about the old classes, we propose a semantic-guided prior that uses old model predictions to discover more precise object boundaries for the new ones.

Concretely, at step $t$ and using the old model $(E \circ F)^{t-1}$, for each pixel $\mathbf{x}_i^t$ we assign the most probable class label $y_i^* = \arg\max_{c \in \mathcal{Y}^{t-1}} \tilde{y}_i^c$ from old classes, yielding the label map $\mathbf{y}^*$. Then, given the set of ground truth image-level labels $\mathcal{L}(\mathbf{x}^t) = \{c | \mathbf{l}_c^t = 1\}$ associated with image $\mathbf{x}^t$, we estimate a similarity map $\mathbf{s}^c$ between each class $l^c$ in $\mathcal{L}(\mathbf{x}^t)$ and the predicted label map $\mathbf{y}^*$:

$$\mathbf{s}^c = \{\mathbf{S}_\Omega \left(\omega(y_i^*), \omega(l^c)\right)\}_{i \in |\mathcal{I}|} \tag{1}$$

where $\omega(c)$ is a vectorial embedding of the semantic class $c$ in a semantic embedding space $\Omega$ and $\mathbf{S}_\Omega$ is a semantic similarity measure defined between the classes in $\Omega$. Different semantic embeddings can be considered, such as Word2Vec (Mikolov et al., 2013), GloVe (Pennington et al., 2014) or BERT (Devlin et al., 2019). These models were trained such that the dot product between embedding vectors, $\mathbf{S}_\Omega$, reflects the semantic similarity between given words. For example in Fig. 1, $\mathbf{S}_\Omega(\omega(\text{'sheep'}),\omega(\text{'cow'})) \gg \mathbf{S}_\Omega(\omega(\text{'sheep'}),\omega(\text{'bkg'}))$, as 'sheep' lies closer to 'cow' in the semantic space than 'background'. In this work, we use BERT (Devlin et al., 2019) for all experiments (see comparisons with other embeddings in Supplementary).

Since one of our aims is to rely on the boundaries of semantically similar classes, the background class plays a crucial role. To ensure not to alter the original predictions made on the background class, we normalize the similarity map such that the score for the '*bkg*' class is equal to 1:

$$s_i^c = \frac{\exp(S_\Omega(\omega(y_i^*),\omega(l^c))/\tau)}{\exp(S_\Omega(\omega(\text{'bkg'}),\omega(l^c))/\tau)} \tag{2}$$

$\tau$ is a scaling hyperparameter. By exploiting such similarity maps we can convert the image labels $l^c$ into pixel-level label maps $\mathbf{s}^c$ (see Figs. 1 and 2).

The key question is, *how to exploit these similarity maps to improve the learning of new classes?* One might be tempted to use Eq. (2) as the only supervisory signal to learn the new classes: this would be sub-optimal, since the learning algorithm also needs to handle cases for which old and new classes are highly dissimilar, or for which the new class region is predicted as background.

To overcome this, we build on the weakly supervised SIS literature, which typically relies on a *localizer* module trained with weak annotations. Its role is to select regions for each semantic class. It is often based on classification activation maps (CAM), which produce discriminative regions for each class that are then used as (pseudo) pixel-level annotations (Zhou et al., 2016a; Araslanov & Roth, 2020). Yet, these methods often fail to provide well-defined maps. To overcome this, the localizer is combined with label propagation (Ahn & Kwak, 2018; Huang et al., 2018) or a CRF (Kolesnikov & Lampert, 2016). Here, we assume a strongly supervised based model, so we can rely instead on the semantic similarity maps defined above. We argue that they can provide a valuable supervisory signal for the localizer. In concrete terms, we define the following loss:

$$\mathcal{L}_{\text{RaSP}}(\mathbf{z},\mathbf{s}) = -\frac{1}{|\mathcal{C}^t||\mathcal{I}|}\sum_{i\in\mathcal{I}}\sum_{c\in\mathcal{C}^t}\sigma(s_i^c)\log(\sigma(z_i^c)) + (1-\sigma(s_i^c))\log(1-\sigma(z_i^c)) \tag{3}$$

where $z_i^c$ is the value assigned by the localizer for class $c$ at pixel $i$, $\sigma$ is the sigmoid function. Given a generic loss for a localizer $\mathcal{L}_{\text{CLS}}$ (an instance of this loss will be detailed in the next section), we can combine the two terms as $\mathcal{L} = \mathcal{L}_{\text{CLS}} + \lambda\mathcal{L}_{\text{RaSP}}$. Intuitively, our proposed loss serves as a regularizer that encourages forward transfer from the old classes to the new ones.

We call our approach **RaSP**, which stands for **R**elation-**a**ware **S**emantic-**P**ior for the WSCIS task as it uses the semantic relationships between the old and the new classes as a prior in the semantic loss. This approach is generic and can be combined with any WSCIS method. In the next section we show how to combine it with the current state-of-the-art approach.

## 3.2 Full integration of RaSP

Without loss of generality, we implement our RaSP loss on top of the WILSON framework (Cermelli et al., 2022). We chose WILSON since it is state of the art and since it relies on a localizer module (Araslanov & Roth, 2020) to tackle WSCIS and, hence, is a good fit to test our loss.

**Background.** WILSON is an end-to-end method for WSCIS that incrementally learns to segment new classes with the supervision of maps generated by a localizer trained with image-level supervision. More specifically, at step t, WILSON is composed of a shared encoder $E^t$ and a segmentation head $F^t$ – which are both incrementally updated – and a localizer head $G^t$, trained from scratch for every task. It also stores a copy of the model from the previous task, $(E \circ F)^{t-1}$.

Given an image $\mathbf{x}$ from the current task, $\tilde{\mathbf{y}} = \sigma((F\circ E)^{t-1}(\mathbf{x})) \in \mathbb{R}^{\mathcal{I}\times|\mathcal{Y}^{(t-1)}|}$ is the output produced by the old model. The scores obtained by the localizer, $\mathbf{z} = (G \circ E)^t(\mathbf{x}) \in \mathbb{R}^{\mathcal{I}\times|\mathcal{Y}^t|}$, are aggregated into a one-dimensional vector $\hat{\mathbf{y}} \in \mathbb{R}^{|\mathcal{Y}^t|}$. Each per-class aggregated score $\hat{y}_c$ is obtained using

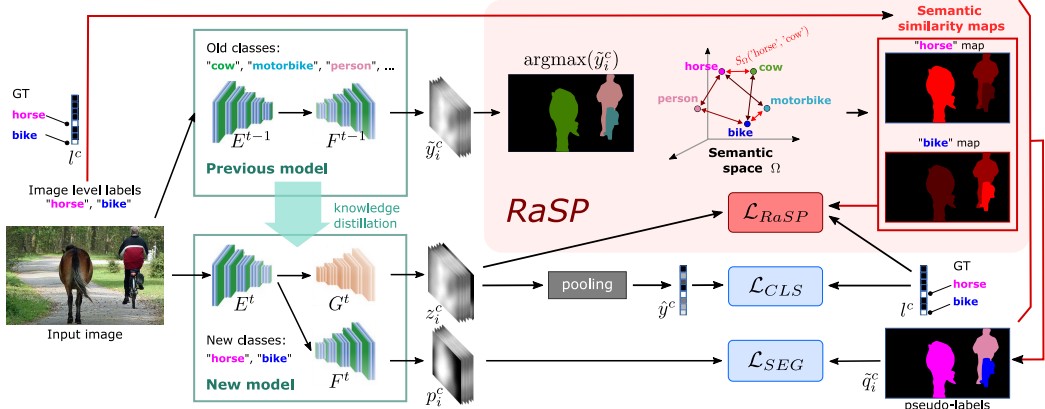

Figure 2: **The RaSP framework.** At training step $t$, the previous model $(E \circ F)^{t-1}$ has been trained to segment cows, motorbikes, and persons among other classes. Presented with an image labeled as containing a horse and a bike, the two new classes to learn, the old model almost perfectly segments out the horse and the bike, except that their pixels are predicted as 'cow' and 'motorbike', respectively. We leverage the semantic relationships between old and new class names to produce similarity maps for the new classes, thus converting the image-level to pixel-level (pseudo) labels.

normalized "Global Weighted Pooling" combined with a focal penalty term (see Supplementary). The score $\hat{y}_c$ can be seen as the likelihood for image $x$ to contain semantic class $c$. This allows training the model with image-level labels using the multi-label soft-margin loss:

$$\mathcal{L}_{\text{CLS}}(\hat{\mathbf{y}}, \mathbf{1}) = -\frac{1}{|\mathcal{C}^t|} \sum_{c \in \mathcal{C}^t} l^c \log(\sigma(\hat{y}^c)) + \sum_{c \in \mathcal{C}^t} (1 - l^c) \log(1 - \sigma(\hat{y}^c)) \tag{4}$$

Note that, although the localizer outputs a $|\mathcal{Y}^t|$-dimensional vector, at task $t$ we are only provided with images and their image-level annotations for the new classes. Therefore, the sum in Eq. (4) is computed only over the new classes. In order to train the localizer for the old classes and prevent the encoder from shifting towards the new classes and forgetting the old ones, Cermelli et al. (2022) propose to distill knowledge from the old model, by adding two knowledge distillation losses. The first one, $\mathcal{L}_{\text{KDE}}$, computes the mean-squared error between the features extracted by the current encoder $E^t$ and those extracted by the previous one $E^{t-1}$. The second distillation loss $\mathcal{L}_{\text{KDL}}$ encourages consistency between the pixel-wise scores for old classes predicted by the localizer $(E \circ G)^t$ and those predicted by the old model $(E \circ F)^{t-1}$ (see details in Supplementary).

Finally, WILSON combines the localizer output with the old model to generate the pseudo-supervision scores $\tilde{q}^c$ that are used to update the segmentation module $(E \circ F)^t$, following

$$\mathcal{L}_{\text{SEG}}(\hat{\mathbf{p}}, \tilde{\mathbf{q}}) = -\frac{1}{|\mathcal{Y}^t||\mathcal{I}|} \sum_{i \in \mathcal{I}} \sum_{c \in \mathcal{Y}^t} \tilde{q}_i^c \log(\sigma(p_i^c)) + (1 - \tilde{q}_i^c) \log(1 - \sigma(p_i^c)) \tag{5}$$

where $\hat{\mathbf{p}} = (E \circ F)^t(\mathbf{x})$ are the predictions from the main segmentation head and $\tilde{\mathbf{q}}$ is the supervisory signal containing i) the old model's predictions for the old classes, ii) the localizer's refined scores for the new classes (see Supplementary for more details) and iii) the minimum between the old model and the localizer scores for the background. The final objective optimized by WILSON is the non-weighted sum of the different loss terms defined above, $\mathcal{L}_{\text{W}} = \mathcal{L}_{\text{CLS}} + \mathcal{L}_{\text{KDL}} + \mathcal{L}_{\text{KDE}} + \mathcal{L}_{\text{SEG}}$.

**Extending WILSON with RaSP.** Since WILSON exploits a localizer-based approach designed for weakly supervised SIS, it constitutes a good starting point to integrate and test our proposed semantic prior. Therefore, we complement WILSON's losses with our loss introduced in Eq (3). Our semantic prior works seamlessly with WILSON's localizer without the need for any ad-hoc architectural changes. Eq. (3) simply requires as input i) the output from the localizer $\mathbf{z} = (E \circ G)^t(\mathbf{x})$ and ii) the semantic similarity maps between new and old classes, obtained via Eqs. (1) and (2). Concretely, with such definitions, our prior loss can also be applied together with WILSON losses, and we can simply optimize the joint loss $\mathcal{L}_{\text{J}} = \mathcal{L}_{\text{W}} + \lambda \mathcal{L}_{\text{RaSP}}$. The hyperparameter $\lambda$ controls the strength of our prior loss, which acts as a regularizer fostering forward transfer from the old to the new classes.

| Input | GT | $(F \circ E)^{t-1}(\mathbf{x}_t)$ | $\mathbf{s^{l_t}}$ | RaSP | WILSON |
|---|---|---|---|---|---|

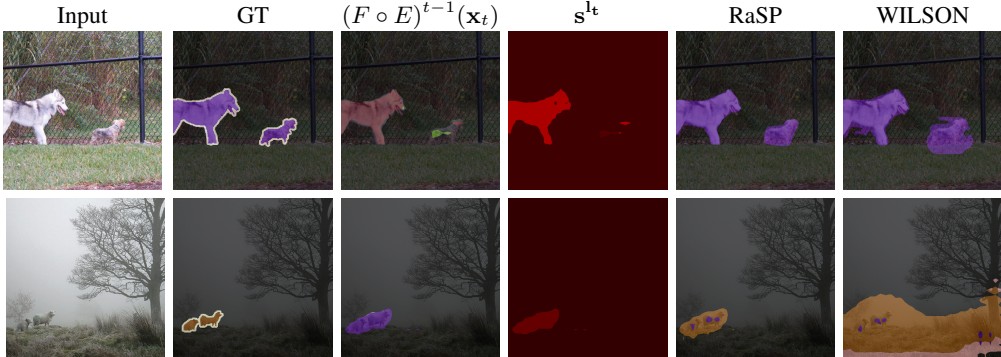

Figure 3: **Visualizations.** Qualitative figures from the *multi-step* **overlap** incremental protocol on 10-2 VOC. From left to right: input image, GT segmentation overlayed, predicted segmentation from old model, semantic similarity map corresponding to the image label (dog / sheep) computed between this label and old classes, predicted segmentation obtained with RaSP and with WILSON. Semantic similarity maps displayed in OpenCV colormap HOT (low ▬▬▬▬ high similarity).

## 4 EXPERIMENTS

**Implementation details.** To be comparable with prior work (Cermelli et al., 2022), we use DeeplabV3 (Chen et al., 2017) as SIS network with either ResNet-101 (He et al., 2016) (for VOC) or Wide-ResNet-38 (Wu et al., 2019) (for COCO). Following WILSON, the localizer is composed of 3 convolutional layers, interleaved with BatchNorm (Ioffe, 2021) and Leaky ReLU layers. For each step, we train the model with SGD for 40 epochs using a batch size of 24. Since the localizer can produce noisy outputs in early training, we do not use $\mathcal{L}_{\text{SEG}}$ for the first 5 epochs (the DeeplabV3 weights remain frozen). In Eq.2, $S_\Omega$ is the dot product and $\omega(y)$ is L2 normalized. We set $\tau = 5$ and $\lambda = 1$ and follow the values suggested by Cermelli et al. (2022) for all other hyperparameters. See Supplementary for different values for $\tau$ and $\lambda$.

**Incremental settings.** Our experiments consider several incremental learning scenarios. We name experiments following the notation $N_b$-$N_t$ to indicate that we first learn with *pixel-level supervision* from $N_b$ base classes, and then sets of $N_t$ *new* classes at a time, with *image-level supervision* only. Given a total number of classes $N$, the number of tasks is $(N - N_b)/N_t + 1$.

*Single-step* settings consider only one incremental learning phase. For instance, in the **15-5 setting** (see Tab. 1), we first train the model on the 15 base classes and then learn the remaining 5 new classes in a single incremental step (bringing the total number of classes to 20). *Multi-step* settings add new classes to the model in multiple sequential steps. The **10-2** setting, for instance, considers 10 base classes and 5 incremental steps which learn 2 new classes at a time. In each table, we indicate results for base classes as 1-$N_b$ and for the new ones as $(N_b + 1)$-$N$.

Each incremental setting can be designed in two ways: i) *Overlap*, if all training images for a given step contain at least one instance of a new class, but they can also contain previous or even future classes; ii) *Disjoint*, if each step consists of images containing only new or previously seen classes, but never future classes. In both protocols, annotations are available for the *new classes only*. We argue that a multi-step protocol with overlap is the most realistic setting. That said, we also consider some other options, especially when it facilitates comparison with prior work.

**Datasets.** Following Cermelli et al. (2022), we experiment mostly with the Pascal VOC dataset (Everingham et al., 2010). VOC consists of $10,582$ training and $1,449$ validation images covering 20 semantic categories. We divide the 20 categories into *base* and *new* classes according to the different protocols described above. We further consider the COCO-to-VOC protocol, where the base classes are the 60 categories from the COCO dataset Lin et al. (2014) (which has 164k training and 5k validation images from 80 semantic categories) that are not available in VOC. In the incremental steps, we learn new classes from VOC categories – following the same protocol as above (*e.g.*, **60-5** is a 5-step protocol where the 20 VOC classes are learned in 4 incremental steps). In this case, in each step we evaluate on the validation sets of both COCO and VOC.

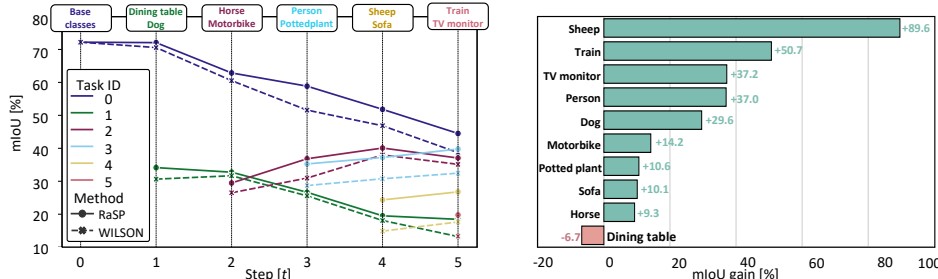

Figure 4: **Left:** Per-task and per-step mIoU for the 10-2 VOC *multi-step* **overlap** incremental setting. **Right:** Per class gain/drop of RaSP w.r.t. WILSON, evaluated for each class in the step it was learned

| Method | Supervision | 15-5 (2 tasks) | | | 10-10 (2 tasks) | | |
| --- | --- | --- | --- | --- | --- | --- | --- |
| | | 1-15 | 16-20 | All | 1-10 | 11-20 | All |
| Fine-Tuning | Pixel | 12.5 | 36.9 | 18.3 | 7.8 | 58.9 | 32.1 |
| LWF (Li & Hoiem, 2016) | Pixel | 67.0 | 41.8 | 61.0 | 70.7 | 63.4 | 67.2 |
| PLOP (Douillard et al., 2021) | Pixel | 75.7 | 51.7 | 70.1 | 69.6 | 62.2 | 67.1 |
| SDR (Michieli & Zanuttigh, 2021b) | Pixel | 75.4 | 52.6 | 69.9 | 70.5 | 63.9 | 67.4 |
| RECALL (Maracani et al., 2021) | Pixel | 67.7 | 54.3 | 65.6 | 66.0 | 58.8 | 63.7 |
| CAM (Zhou et al., 2016b) | Image | 69.9 | 25.6 | 59.7 | 70.8 | 44.2 | 58.5 |
| SEAM (Wang et al., 2020) | Image | 68.3 | 31.8 | 60.4 | 67.5 | 55.4 | 62.7 |
| SS (Araslanov & Roth, 2020) | Image | 72.2 | 27.5 | 62.1 | 69.6 | 32.8 | 52.5 |
| EPS (Lee et al., 2021) | Image | 69.4 | 34.5 | 62.1 | 69.0 | 57.0 | 64.3 |
| WILSON (Cermelli et al., 2022) | Image | 74.2 | 41.7 | 67.2 | 70.4 | 57.1 | 65.0 |
| WILSON† (Cermelli et al., 2022) | Image | **76.3** | 44.1 | 69.3 | 71.4 | 56.1 | 64.9 |
| RaSP (Ours) | Image | 76.2 | **47.0** | **70.0** | **72.3** | **57.2** | **65.9** |
| | | (↓0.1%) | (↑6.6%) | (↑1.0%) | (↑1.3%) | (↑1.6%) | (↑1.5%) |

| | | 10-5 (3 tasks) | | | 10-2 (6 tasks) | | |
| --- | --- | --- | --- | --- | --- | --- | --- |
| | | 1-10 | 11-20 | All | 1-10 | 11-20 | All |
| WILSON† (Cermelli et al., 2022) | Image | 66.8 | 46.5 | 58.1 | 38.7 | 22.4 | 32.5 |
| RaSP (Ours) | Image | **68.8** | **49.1** | **60.4** | **44.5** | **28.4** | **38.6** |
| | | (↑3.0%) | (↑5.6%) | (↑4.0%) | (↑15.0%) | (↑26.8%) | (↑18.8%) |

Table 1: **VOC results.** The mIoU (in %) scores for both *single-step* (2 tasks, top) and *multi-step* (bottom) **overlap** incremental settings on VOC. For each experiment, the three different columns indicate performance on base, new and all 21 classes (including background), respectively. For RaSP (Ours), we further report the relative gain/drop in performance (in %) w.r.t. WILSON†.

**Evaluation metrics.** We evaluate all models using the standard mean Intersection over Union (mIoU) (Everingham et al., 2010). For all scenarios in Tabs. 1-3, we report the mIoU scores evaluated after the last incremental step. Each time we report 3 values: for the base task (considering results on the base classes excluding the background), for the subsequent ones (new classes added during the incremental steps) and finally considering all the classes including the background (All).

## 4.1 MAIN RESULTS

**Comparison to the state of the art.** We compare our proposed RaSP with several state-of-the-art class-incremental learning methods that use either pixel-level or image-level annotations in the incremental steps. We mainly focus on WSCIS methods, which allow fair comparisons. Pixel-supervised methods are interesting but not comparable as they use a prodigious amount of extra-supervision. The best performing method with image-level and pixel-level supervision are respectively bolded and underlined in Tables. Since Cermelli et al. (2022) tested WILSON only for single-step incremental settings, we ran experiments in the other settings using the official implementation provided by the authors (`https://github.com/fcdl94/WILSON`). For comparability, we also re-ran experiments on single-task settings. "WILSON†" indicates our reproduced results while "WILSON" corresponds to the original numbers from the paper. We further report in tables the relative gain/drop in performance (in %) of our RaSP w.r.t. WILSON†, within brackets.

**Results.** In Tab. 1, we show results for both single-step and multi-step incremental settings, on VOC, using the *overlap* protocol. We observe that our RaSP outperforms WILSON in almost all the considered settings. In particular, the relative gain (in %) w.r.t. WILSON grows wider as the number

| Method | Supervision | 60-20 (2 tasks) COCO | | | VOC |
|---|---|---|---|---|---|
| | | 1-60 | 61-80 | All | All |
| Fine-Tuning | Pixel | 1.9 | 41.7 | 12.7 | 75.0 |
| LWF (Li & Hoiem, 2016) | Pixel | 36.7 | 49.0 | 40.3 | 73.6 |
| ILT (Michieli & Zanuttigh, 2019) | Pixel | 37.0 | 43.9 | 39.3 | 68.7 |
| PLOP (Douillard et al., 2021) | Pixel | 35.1 | 39.4 | 36.8 | 64.7 |
| CAM (Zhou et al., 2016b) | Image | 30.7 | 20.3 | 28.1 | 39.1 |
| SEAM (Wang et al., 2020) | Image | 31.2 | 28.2 | 30.5 | 48.0 |
| SS (Araslanov & Roth, 2020) | Image | 35.1 | 36.9 | 35.5 | 52.4 |
| EPS (Lee et al., 2021) | Image | 34.9 | 38.4 | 35.8 | 55.3 |
| WILSON (Cermelli et al., 2022) | Image | 39.8 | **41.0** | 40.6 | **55.7** |
| WILSON† (Cermelli et al., 2022) | Image | **41.1** | **41.0** | **41.6** | 54.8 |
| RaSP (Ours) | Image | **41.1** (0.0%) | 40.7 (↓0.7%) | **41.6** (0.0%) | 54.4 (↓0.7%) |

| | | 60-5 (5 tasks) COCO | | | VOC | 60-2 (11 tasks) COCO | | | VOC |
|---|---|---|---|---|---|---|---|---|---|
| | | 1-60 | 61-80 | All | All | 1-60 | 61-80 | All | All |
| WILSON† (Cermelli et al., 2022) | Image | 30.1 | 28.0 | 30.2 | **42.0** | 10.2 | 14.8 | 12.2 | 24.1 |
| RaSP (Ours) | Image | **33.0** (↑9.6%) | **28.2** (↑0.7%) | **32.5** (↑7.6%) | 41.7 (↓0.7%) | **14.6** (↑43.1%) | **16.5** (↑11.5%) | **15.9** (↑30.3%) | **26.9** (↑11.6%) |

Table 2: **COCO-to-VOC results.** The mIoU (in %) scores for both *single-step* (2 tasks, top) and *multi-step* (bottom) **overlap** incremental settings on **COCO-to-VOC**. For each experiment, the first three columns indicate performance on base, new and all classes (81 including background) computed on COCO, respectively; last column indicates performance on all classes for VOC.

of incremental steps increases, with RaSP achieving +26.8% relative improvement over WILSON in new class performance, in the 10-2 setting. Not only our semantic-prior loss improves new class performance but also it leads to 15% lesser forgetting w.r.t. WILSON. We provide a few qualitative examples in Fig. 3, showing how the semantic maps aid the final segmentation.

Fig. 4 (left) shows the mIoU scores per task and per step for the 10-2 VOC *overlap* setting, indicating which classes are learned at each step (for WILSON and RaSP). This plot shows how our method consistently improves over WILSON throughout the learning sequence. In Fig. 4 (right), we report RaSP's per-class relative percentage improvement w.r.t. WILSON, computed at each step.

We show results for the COCO-to-VOC benchmark (*overlap* protocol) in Tab. 2. RaSP performs comparably with WILSON in the 2-task setting, but outperforms it when more increments and fewer classes per increment are considered – from 60-5 (5 tasks) to 60-2 (11 tasks). We observe both improvements for new classes and reduced forgetting on the old ones.

## 4.2 LEARNING ONE NEW CLASS AT A TIME

A limitation of state-of-the-art approaches for WSCIS is their underwhelming behavior when learning one new class at a time: the model fails to learn and undergoes drastic forgetting. This is due to the fact that Eq. (4) is optimized for a single positive class: the lack of negative classes leads to gross overestimation of the foreground, causing the localizer to provide poor pseudo-labels to the segmentation head, with a negative effect on old classes as well. We show in Tab. 3 (top-half) results of WILSON and RaSP for two single-class incremental settings (15-1 and 10-1), using VOC. Both methods struggle with learning new classes, yielding poor performance compared to pixel-supervised methods. These fully-supervised methods can learn the new classes better since their annotations are composed of both positive foreground-object pixels and negative background pixels.

**A solution: episodic memory.** To circumvent this issue we store a small number of images from base classes in a fixed-size memory $\mathcal{M}$. Intuitively, samples in memory help the localizer by providing negative samples. We show in Tab. 3 (bottom-half) that storing as little as 100 past samples from the dataset significantly improves learning new classed for both WILSON and RaSP, with RaSP + $\mathcal{M}$ outperforming WILSON + $\mathcal{M}$ (28.3% *vs* 20.8% in the 15-1 setting). Unsurprisingly, it also helps retaining performance on the base classes. Similar observations hold for the 10-1 setting.

**External data as an alternative.** Inspired by RECALL (Maracani et al., 2021), we consider the option of retrieving samples for base classes from external sources. We define this memory as $\mathcal{M}_{\text{ext}}$. Concretely, we retrieve 100 samples per base class from ImageNet (by creating a mapping with VOC

| | Method | Supervision | 15-1 (6 tasks) | | | 10-1 (11 tasks) | | |
|---|---|---|---|---|---|---|---|---|
| | | | 1-15 | 16-20 | All | 1-10 | 11-20 | All |
| w/o memory | ILT (Michieli & Zanuttigh, 2019) | Pixel | 4.9 | 7.8 | 5.7 | 16.5 | 1.0 | 9.1 |
| | MiB (Cermelli et al., 2020) | Pixel | 35.1 | 13.5 | 29.7 | 15.1 | 14.8 | 15.0 |
| | WILSON† (Cermelli et al., 2022) | Image | 0.0 | 2.3 | 0.6 | 0.0 | 0.2 | 0.1 |
| | RaSP (Ours) | Image | 17.7 | 0.9 | 13.2 | 2.0 | 0.7 | 1.3 |
| w/ memory | WILSON† + $\mathcal{M}$ | Image | 61.5 | 20.8 | 52.5 | 33.4 | 24.6 | 30.0 |
| | RaSP (Ours) + $\mathcal{M}$ | Image | 63.3 | 28.3 | 56.0 | 38.9 | 30.9 | 36.9 |
| | WILSON† + $\mathcal{M}_{ext}$ | Image | **75.7** | 32.9 | 65.9 | **66.8** | 34.9 | 52.3 |
| | RaSP (Ours) + $\mathcal{M}_{ext}$ | Image | **75.7** | **35.2** | **66.6** | **66.8** | **39.1** | **54.4** |
| | RECALL (Web) (Maracani et al., 2021) | Pixel | 67.8 | 50.9 | 64.8 | 65.0 | 53.7 | 60.7 |

Table 3: **Effect of memory.** Results on single-class *multi-step* **overlap** incremental setting on VOC. $\mathcal{M}$ and $\mathcal{M}_{ext}$ indicate memories of previously seen or external samples, respectively.

| Method | Supervision | VOC (5-shot) | | | VOC (2-shot) | | | COCO (5-shot) | | | COCO (2-shot) | | |
|---|---|---|---|---|---|---|---|---|---|---|---|---|---|
| | | 1-15 | 16-20 | HM | 1-15 | 16-20 | HM | 0-60 | 61-80 | HM | 0-60 | 61-80 | HM |
| Fine-Tuning | Pixel | 55.8 | 29.6 | 38.7 | 59.1 | 19.7 | 29.5 | 41.6 | 12.3 | 19.0 | 41.5 | 7.3 | 12.4 |
| WI (Qi et al., 2018) | Pixel | 63.3 | 21.7 | 32.3 | 63.3 | 19.2 | 29.5 | 43.6 | 8.7 | 14.6 | 44.2 | 7.9 | 13.5 |
| AMP Siam et al. (2019) | Pixel | 51.9 | 18.9 | 27.7 | 54.4 | 18.8 | 27.9 | 34.6 | 11.0 | 16.7 | 35.7 | 8.8 | 14.2 |
| MiB (Cermelli et al., 2020) | Pixel | 65.0 | 28.1 | 39.3 | 63.5 | 12.7 | 21.1 | 44.7 | 11.9 | 18.8 | 44.4 | 6.0 | 10.6 |
| PIFS (Cermelli et al., 2021) | Pixel | 60.0 | 33.3 | 42.8 | 60.5 | 26.4 | 36.8 | 42.8 | 15.7 | 23.0 | 40.9 | 11.1 | 17.5 |
| WILSON† (Cermelli et al., 2022) | Image | 64.1 | 20.5 | 31.1 | 63.3 | 10.2 | 17.6 | 45.0 | **5.8** | **10.3** | **43.6** | 1.9 | 3.6 |
| RaSP | Image | **64.4** | **21.3** | **32.0** | **63.5** | **10.7** | **18.3** | **45.1** | 5.6 | 10.0 | 43.5 | **2.0** | **3.8** |
| | | (↑0.5%) | (↑3.9%) | (↑2.9%) | (↑0.3%) | (↑4.9%) | (↑4.0%) | (↑0.2%) | (↓3.4%) | (↓2.9%) | (↓0.2%) | (↑5.3%) | (↑5.6%) |

Table 4: **Few-shot results.** The mIoU (in %) scores for the *single-step* (2 tasks) incremental few-shot SiS settings on the PASCAL-$5^i$ and COCO-$20^i$ benchmarks, for 5-shot and 2-shot cases. We show the average results over the 4 folds as in (Cermelli et al., 2021). For each experiment, columns report performance on old classes, new classes, and the Harmonic-Mean (HM) of the two scores.

classes). This further improves both WILSON + $\mathcal{M}_{ext}$ and RaSP + $\mathcal{M}_{ext}$ compared to the previous episodic memory. RECALL performs better on new classes, but i) relies on pixel-level supervision and ii) uses significantly more web-crawled images; therefore, it is not directly comparable.

### 4.3 CLASS-INCREMENTAL FEW-SHOT SEGMENTATION

We compare RaSP and WILSON on the task of Incremental Few-Shot Segmentation (iFSS) (Cermelli et al., 2021), where the model learns incrementally from only few images. This is a challenging setting, only tested so far with pixel-level supervision. Here, we add the challenging constraints that the new training images are only weakly annotated. Following Cermelli et al. (2021), we consider 4 folds of 5 classes for PASCAL-$5^i$ and the 4 folds of 20 classes for COCO-$20^i$. where each fold in turn is used as incremental setting with the other classes forming the base task.

Tab. 4 reports average results over the 4 folds per-fold (per-fold results in Supplementary). Bottom lines contain results obtained by WILSON and RaSP; as expected, in the case of COCO-$20^i$ both methods perform poorly, which is not surprising as even the strongly supervised methods (top lines) have difficulties to learn the new classes. On the other hand, on PASCAL-$5^i$, not only RaSP consistently outperforms WILSON, but in the 5-shot case it also outperforms or performs on par with some of the strongly supervised methods. Finally, we can appreciate that the performance of RaSP on the base classes remains comparable or often outperforms most of the strongly supervised methods, where the higher performance on the new classes tends to come with a more severe forgetting.

### 5 CONCLUSIONS

We proposed a new method for Weakly Supervised Class-Incremental Semantic Image Segmentation. Motivated by the observation that new classes to be added to the model often bare resemblance with the old ones that it already knows, we designed a Relation-aware Semantic Prior loss (RaSP) that fosters forward transfer. It leverages semantic similarity between old and new class names in a loss that guides the learning of new categories. We validated our idea on a broad variety of benchmarks. In particular, we demonstrated that our method is resilient to unexplored and challenging scenarios which increase the number of tasks and reduce data availability for new classes.

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
