# OpenReview forum: "Semantic Prior for Weakly Supervised Class-Incremental Segmentation"
_ICLR.cc/2023/Conference — Submitted to ICLR 2023_

### Official Review · Reviewer_RkMT · 2022-10-25

**Confidence:** 3
**Correctness:** 3
**Technical Novelty And Significance:** 2
**Empirical Novelty And Significance:** 3
**Recommendation:** 5

**Clarity, Quality, Novelty And Reproducibility:**

Clarity
The paper does do well in present the core idea of the proposed approach. However, it is not very clear about the key difference between the proposed approach and the one from WILSON.

Quality
The paper is presented in a good quality. The experiments have also demonstrated the effectiveness of the proposed approach in several continuous learning setting.

Novelty
The paper has a incremental contribution on top of the WILSON work.


Reproducibility
The paper seems to be easily reproduced.

**Details Of Ethics Concerns:**

No concerns.

**Strength And Weaknesses:**

Pros

The approach naturally generalized the prior works on image-classification to the weakly semantic image segmentation, which is novel and inspiring. I think approaching the problem from the loss perspective is reasonable.

The proposed approach seems to be working well on the proposed settings. The proposed approach shows good results in the settings of effect of memory and few-shot.

Cons

The proposed approach is relative not incremental on top of the WILSON project. As the main idea in this paper is to add the RASP loss on top of the WILSON framework. However, it does not seem that the additional RASP loss adds the big advantages to the WILSON framework. More experiments should be provided to justify if the additional useful signals are provided from the localizer. Specifically, we need to see improvements of the slice and dice metric improvements of the new classes once we have applied the RASP loss and use the information from the old class. The Baseline in that setting is that we are not using RASP loss, but just run CRF-like approach.

Another question is about the RASP loss. There is no guarantee that the localizer for old tasks is providing the positive impact to the new tasks, unless we have made some careful choice on the old tasks and new tasks, which is not possible in the real-world use cases. How would one use the proposed approach in practice?

One design choice needs to be justified. In particularly, the proposed approach chooses to treat the new class region as background and build the proposed approach on a localizer module trained with weak annotations. In contrast, there are multiple approaches in literatures in salient region detection that shows promising results in segmenting the background, and plenty of the approaches using super-pixels. It is not clearly why the proposed approach pick up the route, which seems to be bias towards to give better thing-like object instance segmentation. The proposed technical choice might not work well on the stuff-like objects instances, which has extended boundaries. These are missed out in the discussion.

The experiments are mostly focus on the easy datasets such as Pascal and COCO. For the proposed to work, I would like to see how the proposed approach work on a more challenging dataset. The simulated experiments on new and old classes seem to be cherry picked.

**Summary Of The Paper:**

The proposed approach address the problem of class-incremental for weakly supervised semantic image segmentation. The goal of this paper is to leverage the semantic relations among the labels to improve the weakly supervised learning for semantic segmentation. The idea is to generalize the WILSON approach for weakly semantic image segmentation. A new loss relation aware semantic prior loss (RASP) is developed to encourage the forward transfer across classes within the learning sequence. The proposed loss is validated in continuous learning setting.

**Summary Of The Review:**

On the positive side, I think the proposed approach has good spirit as it has shown clear improvements on class-incremental setting for weakly semantic image segmentation. The proposed approach has also touched the drawbacks of the WILSON framework, and added the new loss RASP which encourages the forward transfer across classes within the continuous learning setting. However, the paper does not seem to show clear improvements. The aggregated metric improvements on the benchmarks do not guarantee that the improvements are from the proposed RASP loss. There are other technical design choices are not well justified.

---

> ### Author Response · Authors · 2022-11-18
> **Responses to Reviewer RkMT (4/4)**
>
> >Q4. *The experiments are mostly focus on the easy datasets such as Pascal and COCO. For the proposed to work, I would like to see how the proposed approach work on a more challenging dataset. The simulated experiments on new and old classes seem to be cherry picked.*
>
> **Answer**: We evaluate on Pascal-VOC and MS-COCO because the only existing WSCI method, WILSON, has established them as benchmarks. Moreover, among the WSSS literature (Araslanov & Roth, 2020; Du *et al.*, 2022; Xu *et al.*, 2022; Xie *et al.*, 2022; Chen *et al.*, 2022), which are closely related to WSCI, it is common to evaluate methods on these two datasets. Thus, we stick to the standards set by the relevant research community for weakly supervised segmentation and evaluate accordingly.
>
> The numbers on the Pascal-VOC and MS-COCO benchmarks in the WSCI settings are far from saturated and therefore cannot be aptly characterized as ''*easy*''. For instance, in the extremely challenging single-class multi-step incremental settings of 15-1 VOC and 10-1 VOC we can observe that the incremental baselines struggle with very low overall performance. While the suggestion of evaluating on an even more challenging dataset, such as ADE20K, sounds interesting, it comprises *stuff* categories and many small objects, which are indeed difficult to address with the weaker image labels as supervision. That is why the WSSS methods limit themselves to Pascal-VOC and MS-COCO, and ADE20K is only used in the class-incremental semantic segmentation literature that use pixel labels in the incremental steps.
>
> Finally, note that *we did not cherry pick any setting in our work*. Please refer to: (i) the [answer](https://openreview.net/forum?id=0vG8GbuPOH3&noteId=8HtOEfLLi5) to **Q4** of Reviewer **PPnr** for a detailed discussion about the chosen evaluation settings, and (ii) the [answer](https://openreview.net/forum?id=0vG8GbuPOH3&noteId=9ypAGpZgD2) to **Q3** of Reviewer **AMCk** where we show that our improvements are consistent across different randomizations of the class ordering. For transparency we have additionally reported the failure cases in Sec. D of the Supplementary.
>
> **References**:
>
> Nikita Araslanov and Stefan Roth. Single-stage semantic segmentation from image labels. In CVPR, 2020.
>
> Ye Du, Zehua Fu, Qingjie Liu, and Yunhong Wang. Weakly supervised semantic segmentation by pixel-to-prototype contrast. In CVPR, 2022.
>
> Lian Xu, Wanli Ouyang, Mohammed Bennamoun, Farid Boussaid, and Dan Xu. Multi-class token transformer for weakly supervised semantic segmentation. In CVPR, 2022.
>
> Jinheng Xie, Xianxu Hou, Kai Ye, and Linlin Shen. Clims: Cross language image matching for weakly supervised semantic segmentation. In CVPR, 2022.
>
> Qi Chen, Lingxiao Yang, Jian-Huang Lai, and Xiaohua Xie. Self-supervised image-specific prototype exploration for weakly supervised semantic segmentation. In CVPR, 2022.

---

> ### Author Response · Authors · 2022-11-18
> **Responses to Reviewer RkMT (3/4)**
>
> > Q3. *One design choice needs to be justified. In particularly, the proposed approach chooses to treat the new class region as background and build the proposed approach on a localizer module trained with weak annotations. In contrast, there are multiple approaches in literatures in salient region detection that shows promising results in segmenting the background, and plenty of the approaches using super-pixels. It is not clearly why the proposed approach pick up the route, which seems to be bias towards to give better thing-like object instance segmentation. The proposed technical choice might not work well on the stuff-like objects instances, which has extended boundaries. These are missed out in the discussion.*
>
> **Answer**: We acknowledge that we are aware of the methods that use saliency maps as additional input for the weakly supervised segmentation algorithms. We have indeed cited one such work, EPS (Lee *et al.*, 2021) in our paper and have compared to it in the main results. Note that saliency detectors first need to be trained on an external and relevant dataset to be useful for segmenting out the foreground from the background. This would add complexity to the overall pipeline. In this work we dispose of the need to be reliant on an external model, and show that an older version of the very same segmentation model can provide essential cues about the location of the new classes. This is advantageous in general because it greatly reduces engineering effort.
>
> We agree that we should have included the discussion about the *stuff* categories. We do not deviate from the scope of the standard evaluation in weakly supervised semantic segmentation (WSSS) tasks. In particular, the state-of-the-art WSSS methods (Araslanov & Roth, 2020; Du *et al.*, 2022; Xu *et al.*, 2022; Xie *et al.*, 2022; Chen *et al.*, 2022) only report performance on the *things* categories of Pascal-VOC and MS-COCO. Due to the nature of weak image-label annotations, the WSSS methods are not able to segment stuff-like instances that do not conform to regular boundaries. Given that we learn the incremental steps with image labels, we also do not include the *stuff* categories in the evaluation, adhering to the relevant literature.
>
> **References**:
>
> Seungho Lee, Minhyun Lee, Jongwuk Lee, and Hyunjung Shim. Railroad is not a train: Saliency as pseudo-pixel supervision for weakly supervised semantic segmentation. In CVPR, 2021.
>
> Nikita Araslanov and Stefan Roth. Single-stage semantic segmentation from image labels. In CVPR, 2020.
>
> Ye Du, Zehua Fu, Qingjie Liu, and Yunhong Wang. Weakly supervised semantic segmentation by pixel-to-prototype contrast. In CVPR, 2022.
>
> Lian Xu, Wanli Ouyang, Mohammed Bennamoun, Farid Boussaid, and Dan Xu. Multi-class token transformer for weakly supervised semantic segmentation. In CVPR, 2022.
>
> Jinheng Xie, Xianxu Hou, Kai Ye, and Linlin Shen. Clims: Cross language image matching for weakly supervised semantic segmentation. In CVPR, 2022.
>
> Qi Chen, Lingxiao Yang, Jian-Huang Lai, and Xiaohua Xie. Self-supervised image-specific prototype exploration for weakly supervised semantic segmentation. In CVPR, 2022.

---

> ### Author Response · Authors · 2022-11-18
> **Responses to Reviewer RkMT (2/4)**
>
> > Q2. *Another question is about the RASP loss. There is no guarantee that the localizer for old tasks is providing the positive impact to the new tasks, unless we have made some careful choice on the old tasks and new tasks, which is not possible in the real-world use cases. How would one use the proposed approach in practice?*
>
> **Answer**: Thank you for bringing up the discussion about an *edge-case* where all the base classes are extremely dissimilar to the new classes. We argue that in a real-world use case it would be quite the contrary as annotations for the entry-level categories are always readily available from existing segmentation benchmarks (e.g. Pascal-VOC, MS-COCO). Learning new classes in incremental steps will mainly be comprised of images from categories that are long tailed or are not commonly annotated at pixel-level. For example, a base model trained on the entry-level class ''bus'' can be used to segment a wide variety of newly manufactured classes of other vehicles that do not substantially differ in appearance such as ''pick-up trucks'', ''cargo trucks'' and ''fire-trucks'' from image labels. Even if we assume the edge-case of dissimilar base classes, our incremental learner can also exploit the semantic similarity between the new classes form the recent past with the later new classes.
>
> In other words, over an extended period of time the RaSP loss will eventually improve upon WILSON and is not strictly reliant on pixel-level annotations from the base classes. Please refer to the [answer](https://openreview.net/forum?id=0vG8GbuPOH3&noteId=9ypAGpZgD2) to **Q3** of Reviewer **AMCk** for an experiment with different class orderings, where we show that RaSP does not hurt performance in such scenarios.

---

> ### Author Response · Authors · 2022-11-18
> **Responses to Reviewer RkMT (1/4)**
>
> We would like to thank Reviewer RkMT for appreciating the novelty of our proposed method that addresses weakly supervised class-incremental segmentation (WSCI) and the improved results over its competitors in a variety of settings.
>
> >Q1. *The proposed approach is relative not incremental on top of the WILSON project. As the main idea in this paper is to add the RASP loss on top of the WILSON framework. However, it does not seem that the additional RASP loss adds the big advantages to the WILSON framework. More experiments should be provided to justify if the additional useful signals are provided from the localizer. Specifically, we need to see improvements of the slice and dice metric improvements of the new classes once we have applied the RASP loss and use the information from the old class. The Baseline in that setting is that we are not using RASP loss, but just run CRF-like approach.*
>
> **Answer**: We would like to draw the reviewer's attention to the following tasks where we improve over WILSON and restate the corresponding relative percentage gain in favour of our method. For instance, in Tab. 1 gains are (i) +1% in 15-5 VOC, (ii) +1.6% in 10-10 VOC, (iii) +4.0% in 10-5 VOC, (iv) +18.8% in 10-2 VOC, overlapped settings. Similar positive pattern can also be observed in the Supplementary where in Tab. A2 for the VOC disjoint gains are (i) +0.9% in 15-5, (ii) +0.7% in 10-10, (iii) +3.2% in 10-5, and (iv) +19.6% in 10-2. While in the setting of 60-20 COCO-to-VOC, our RaSP is outperformed by WILSON, our method shines when the number of incremental steps increases (e.g., +30.3% and +11.6% for COCO and VOC, in Tab. 2). Note that in real-world applications, where the number of steps will increase over the lifetime of a model, the proposed RaSP improves dramatically over WILSON - making it stand out in terms of realism and effectiveness. Finally, in the hardest scenario of single-class incremental settings (see Tab. 3 and Tab. A3) RaSP consistently outperforms WILSON by significant margins. Considering the totality of the improvements over WILSON we believe that the advantage offered by RaSP is substantial.
>
> In Fig. 4 (right) and Fig. A2 we show the break down of the gain obtained by RaSP with respect to WILSON for each of the new classes. For instance, in Fig. 4 (right) for the multi-step 10-2 VOC setting, it is evident that in some occasions the relative percentage gain in performance for the new classes (*e.g.*, +89.6% for class ''sheep'' and +50.7% for class ''train'') is truly substantial. We have reported the class-wise and step-wise mIoU scores for both WILSON and RaSP in Sec. B.5 of the Supplementary.
>
> We did not report custom baselines that use CRFs because the WILSON internally uses PAMR (Araslanov & Roth, 2020), which is a local mask refinement technique similar to CRF. In details, guided by the same intuition as the CRFs, PAMR ensures that the nearby regions sharing the same appearance are assigned to the same class. However, PAMR achieves the same effect as the CRFs while speeding up the training process. As WILSON's implementation includes such local refinement, we did not consider CRF due to redundancy.
>
> **References**:
>
> Nikita Araslanov and Stefan Roth. Single-stage semantic segmentation from image labels. In CVPR, 2020.

---

### Official Review · Reviewer_AMCk · 2022-10-25

**Confidence:** 4
**Correctness:** 2
**Technical Novelty And Significance:** 2
**Empirical Novelty And Significance:** 2
**Recommendation:** 3

**Clarity, Quality, Novelty And Reproducibility:**

The paper is well-written and easy to follow.

As I stated earlier, the novelty is only marginal: integration of a simple semantic loss into the existing WILSON framework.

The paper seems to be easily reproducible.

**Strength And Weaknesses:**

**Strengths**

+ Weakly supervised class-incremental segmentation is very relevant and important and only a few approaches have been proposed in the literature
+ The idea of using semantic relations between past classes and new classes makes sense.
+ The paper is easy to follow and well-structured.
+ Experiments and comparison to the state of the art are promising.

**Weaknesses**

+ My main concern is related to the novelty of the approach which just consists in adding new semantic losses in the WILSON framework and is thus marginal.
+ Moreover, the choice and design of the semantic losses are very simple and could be more discussed. For instance, how to take into account stronger relations between classes than just embedding similarity ones? For instance, in semantic segmentation part-whole relations or spatial relations between classes could also be of great interest and the proposed approach can not differentiate between the different types of relations.
+ Since the claim of the paper is to use conceptual relations between past and new classes, I would appreciate more experiments focusing on these specific points: for instance a setting with explicit relations between the different incremental steps and various configurations: specialization relations, generalization relations, mereo-topological relations...


**Summary Of The Paper:**

This paper proposes a new loss for weakly supervised (image-level labels) class-incremental segmentation that enables taking into account semantic relations between past classes and new classes. The proposed approach can be integrated to existing semantic image segmentation approaches that used a localizer module. In the paper, they integrate the loss to the recent WILSON framework. Experiments on Pascal VOC and COOC-to-Voc with different incremental settings are promising and show the interest of the approach.



**Summary Of The Review:**

The paper tackles the relevant and important weakly supervised (image-level labels) class-incremental segmentation problem. It proposes to integrate a semantic loss term into an existing framework that improves the state-of-the-art but the novelty of the proposed approach is only marginal and important component of the proposed approach such as the notion of semantic priors is not deeply studied.

---

> ### Author Response · Authors · 2022-11-18
> **Responses to Reviewer AMCk (3/3)**
>
> > Q3. *Since the claim of the paper is to use conceptual relations between past and new classes, I would appreciate more experiments focusing on these specific points: for instance a setting with explicit relations between the different incremental steps and various configurations: specialization relations, generalization relations, mereo-topological relations.*
>
> **Answer**: Thank you for another valuable suggestion, evaluating on additional configurations that differ by the relationship between the old and new classes. Inferring *specialization*, *generalization* and *mereo-topological* relations presents some challenges in the WSCI setting because of the following: (i) since we are confined to the WSCI benchmarks Pascal-VOC (Everingham *et al.*, 2010) and MS-COCO (Lin *et al.*, 2014), which contain per-object entry-level categories as annotations, the model cannot predict a set of attributes associated with a given object. Characteristic and descriptive attributes are quintessential to discover such relationships between the old and the new classes (Iannone et al., 2009); (ii) as an alternative, an external knowledge graph (Wang et al., 2017), which contains descriptive attributes associated with the object names, can indeed be a candidate solution to infer such relationships. Given that the WSCI benchmarks contain a diverse set of images associated with a generically annotated entry-level category (*e.g.*, ''bus'', ''train'', etc.), the use of generic attributes will still be an incomplete representation of the scene described in an image. Due to the limitations inherent to these benchmarks, we resort to a simple embedding similarity, which makes the framework simpler and still effective. To demonstrate that our metric is versatile, we chose the 15-5 VOC setting of 15 base classes and 5 novel classes and randomized the old-novel splits. We ran experiments on four such random splits and report the results in the table below.
>
> | Method |         |  15-5a |     |         | 15-5b|      |         | 15-5c|       |         | 15-5d|      |          | Mean|      |
> | --------- | ----- | ------- | -- | ----- | ------ | --- | ----- | ------ | ---  |----- | ------ | ---  | ----- | ------ | ---  |
> |             | 1-15 | 16-20 | All| 1-15 | 16-20 | All | 1-15 | 16-20 | All | 1-15 | 16-20 | All | 1-15 | 16-20 | All |
> |WILSON| 75.8 | 45.2 | 69.3 | 71.2 | 48.5 | 66.7 | 68.7 | 42.7 | 63.6 | 66.5 | 56.2 | 65.3 | 70.6 | 48.2 | 66.2|
> |RaSP (ours) | **75.9**| **47.5**| **69.9**| **71.8**| **53.3**| **68.4**| **70.8**| **44.5**| **65.5**| **66.7**| **57.8**| **65.9**| **71.3**| **50.8** | **67.4**|
>
> From the table it is evident that RaSP outperforms WILSON on the four randomly chosen base-novel classes split of VOC, denoted by 15-5a, 15-5b, 15-5c and 15-5d, indicating that our improvements are consistent on all of the class orderings. While the improvement by RaSP varies among the base-novel splits, they do not drop below WILSON. Thus we believe that our proposed method is well suited for real world applications where the classes will appear in a random (and *unknown*) order and yet our incremental learner can perform better than its competitors. We have added the results of random splits in Sec. B.4 of the Supplementary.
>
> **References**:
>
> Mark Everingham, Luc Van Gool, Christopher KI Williams, John Winn, and Andrew Zisserman. The pascal visual object classes (voc) challenge. In IJCV, 2010.
>
> Tsung-Yi Lin, Michael Maire, Serge Belongie, James Hays, Pietro Perona, Deva Ramanan, Piotr Doll ́ar, and C Lawrence Zitnick. Microsoft coco: Common objects in context. In ECCV, 2014.
>
> Luigi Iannone, Alan Rector, and Robert Stevens. Embedding knowledge patterns into owl. In European Semantic Web Conference, 2009.
>
> Quan Wang, Zhendong Mao, Bin Wang, and Li Guo. Knowledge graph embedding: A survey of approaches and applications. In IEEE Transactions on Knowledge and Data Engineering, 2017.

---

> ### Author Response · Authors · 2022-11-18
> **Responses to Reviewer AMCk (2/3)**
>
> >Q2. *Moreover, the choice and design of the semantic losses are very simple and could be more discussed. For instance, how to take into account stronger relations between classes than just embedding similarity ones? For instance, in semantic segmentation part-whole relations or spatial relations between classes could also be of great interest and the proposed approach can not differentiate between the different types of relations.*
>
> **Answer**: Thank you for making an interesting remark about going beyond the embedding similarity. We agree with the reviewer that incorporating the part-whole and spatial relationships would be of great interest, and can potentially improve the forward transfer in WSCI. Despite this promise, we consider this setting to be out of the scope of our work because: (i) identifying part-whole relations will require part annotations for each object in the base training set (e.g. Pascal-Part-58, Pascal-Part-108 and ADE20K-Part  (Michieli & Zanuttigh, 2022)), which are not accessible in the WSCI task, and (ii) although it has been shown in the co-part segmentation literature (Siarohin *et al.*, 2021a;b) that part segmentations can be obtained without supervision, such methods rely on modelling motion and need to be trained on curated and object centric single-class videos datasets having static background. Moreover, the discovered part names do not come with associated label names, but just label ids, which cannot contribute towards building *part-of* relationships. Given the limited resources at disposal in the WSCI setting, i.e., object-level pixels labels for the base classes and image labels for the new ones, we do not possess adequate tools for exploring different kind of relations without adding complex training modules. Nevertheless, the simple embedding similarity proposed in our work can already improve forward transfer, which is encouraging.
>
> **References**:
>
> Umberto Michieli and Pietro Zanuttigh. Edge-aware graph matching network for part-based semantic segmentation. In IJCV, 2022
>
> Aliaksandr Siarohin, Subhankar Roy, Stephane Lathuiliere, Sergey Tulyakov, Elisa Ricci, and Nicu Sebe. Motion-supervised co-part segmentation. In ICPR, 2021a.
>
> Aliaksandr Siarohin, Oliver J Woodford, Jian Ren, Menglei Chai, and Sergey Tulyakov. Motion representations for articulated animation. In CVPR, 2021b.

---

> ### Author Response · Authors · 2022-11-18
> **Responses to Reviewer AMCk (1/3)**
>
> We would like to thank Reviewer AMCk for praising the relevance of the weakly supervised class-incremental segmentation (WSCI) task and the validity of our proposed method. We also appreciate that the reviewer found our experimental evaluation and comparison to the state of the art promising.
>
> > Q1. *My main concern is related to the novelty of the approach which just consists in adding new semantic losses in the WILSON framework and is thus marginal.*
>
> **Answer**: Please refer to the [answer](https://openreview.net/forum?id=0vG8GbuPOH3&noteId=zmby_Kw0mJ-) to **Q2** of Reviewer **paY4**.

---

### Official Review · Reviewer_PPnr · 2022-10-25

**Confidence:** 4
**Clarity, Quality, Novelty And Reproducibility:** See above for the evaluation of the c…
**Correctness:** 3
**Technical Novelty And Significance:** 2
**Empirical Novelty And Significance:** Not applicable
**Recommendation:** 3

**Strength And Weaknesses:**

Strengths:
- The problem setup of this work, which requires only weak annotation in incremental semantic segmentation, is useful in practice but less explored.
- The idea of exploring semantic similarity for forward transfer seems novel for the task of weakly-supervised incremental segmentation.
- The paper is mostly well-written and easy to follow.

Weaknesses:
- The overall novelty of the proposed framework is limited. It heavily relies on the existing framework WILSON and the main contribution is an improved WSSS loss integrated within each incremental step.
- The assumption on the background class is problematic for incremental segmentation. This work seems to assume a static background class, which is not true in this task. Due to this background drifting issue, the proposed pseudo label generation strategy seems less effective: In equation (2), the bkg class would have a high similarity to previous bkg class which includes novel classes. This would largely suppress the new classes and conflict with the CAM loss. To make this work, it seems to require a good base segmentation network to generate foreground masks, which is difficult for the incremental setup.
- The experimental results are mixed or marginally improved compared to the WILSON framework. On some task settings of the VOC benchmark, the overall improvements are around 1% and on some task settings of the COCO-to-VOC benchmark, the proposed method is inferior to the baseline.
- The experimental evaluation settings are limited. All of them start from a base learning task with many classes (15 or 60) and incrementally learn a smaller number of classes (5 or 20). It should be evaluated on more challenging settings with fewer base classes and more novel classes. Also, the evaluation on larger datasets in ISS, such as ADE, should be added.

**Summary Of The Paper:**

The paper presents a weakly-supervised incremental semantic segmentation strategy, which starts from a base learning task with fully-annotated images and then incrementally learns novel classes with image-level labels only. To tackle this problem, it introduces a pseudo label generation method for the weakly supervised semantic segmentation (WSSS) at each incremental stage, which infers the pseudo labels of the new classes based on the model prediction of the old classes and the semantic similarity between new & old categories in a word embedding space. The proposed WSSS loss is then integrated into an existing weakly-supervised incremental semantic segmentation framework,  WILSON,  in order to build a full pipeline.  The authors evaluate the method on the PASCAL VOC dataset and COCO-to-VOC benchmark with comparisons to prior methods.

**Summary Of The Review:**

The proposed weakly supervised ISS method seems less convincing due to the limitations on its novelty and pseudo-label generation, mixed performance, and the lack of sufficient experimental evaluation.

---

> ### Author Response · Authors · 2022-11-18
> **Responses to Reviewer PPnr (4/4)**
>
> >Q4. *The experimental evaluation settings are limited. All of them start from a base learning task with many classes (15 or 60) and incrementally learn a smaller number of classes (5 or 20). It should be evaluated on more challenging settings with fewer base classes and more novel classes. Also, the evaluation on larger datasets in ISS, such as ADE, should be added.*
>
> **Answer**: As discussed in Sec. 4, to fairly compare with the existing state-of-the-art WSCI methods, we adopt the exact same class splits for Pascal-VOC and COCO-to-VOC benchmarks as proposed in WILSON by Cermelli et al. (2022). However, WILSON is evaluated in *single-step* incremental settings (15-5 VOC, 10-10 VOC, 60-20 COCO-to-VOC), which in practical applications is of little use. To broaden the experimental settings and simulate the real world application where an incremental learner is expected to learn over a far longer sequence, often times with as few as one single class, we introduced several *multi-step* incremental settings (10-5 VOC, 10-2 VOC, 15-1 VOC, 10-1 VOC, 60-5 COCO-to-VOC and 60-2 COCO-to-VOC). As expected, the overall performance of the WSCI model drops when the sequence of tasks become longer and the number of classes in each task fewer. Finally, we also evaluated on the challenging few-shot WSCI scenario to gauge the performance when only a few image labelled samples are available (Sec. 4.3 of the main and Sec. B.6 of the Supplementary). Given that we go beyond the *limited* existing settings of WILSON, and stress-test the WSCI models in more challenging and practical settings, we believe that our experimental evaluation is adequate and encompasses various incremental scenarios.
>
> Evaluating the model on fewer base and more novel classes is an interesting suggestion, thanks a lot. From our experimental evaluation on all the aforementioned settings, we found that the proposed RaSP improves WILSON in most settings, barring a few odd scenarios (for *e.g.*, 60-20 COCO-to-VOC). We are confident that the additional setting of fewer base and more novel classes will demonstrate similar positive trend for RaSP - as far as new classes share similarities with some of the old ones. Unfortunately, given limited time and compute, we are not able to run such a suite of experiments, but we plan to include them in the future.
>
> Please note that it is fairly common in the WSSS literature (Araslanov & Roth, 2020; Du *et al.*, 2022; Xu *et al.*, 2022; Xie *et al.*, 2022; Chen *et al.*, 2022) to evaluate methods only on the Pascal-VOC and the original MS-COCO (*things*) datasets. While the suggestion of evaluating on the ADE20K sounds interesting, this dataset comprises *stuff* categories (*e.g.*, ''floor'', ''wall'', etc.) and many small objects, which are indeed difficult to address with the weaker image labels as supervision. That is why WSSS methods limit their evaluation to Pascal-VOC and MS-COCO, whereas ADE20K is only used in the class-incremental semantic segmentation literature that uses pixel-level labels in the incremental steps. Thus, we follow suit of the WSSS and WSCI literature and present results on the standard Pascal-VOC and COCO. To summarize, we would like to stress that the aforementioned limitation is intrinsic to *every* WSSS and WSCI method, and by no means just exclusive to our proposed RaSP.
>
> **References**:
>
> Fabio Cermelli, Dario Fontanel, Antonio Tavera, Marco Ciccone, and Barbara Caputo. Incremental Learning in Semantic Segmentation from Image Labels. In CVPR, 2022.
>
> Nikita Araslanov and Stefan Roth. Single-stage semantic segmentation from image labels. In CVPR, 2020.
>
> Ye Du, Zehua Fu, Qingjie Liu, and Yunhong Wang. Weakly supervised semantic segmentation by pixel-to-prototype contrast. In CVPR, 2022.
>
> Lian Xu, Wanli Ouyang, Mohammed Bennamoun, Farid Boussaid, and Dan Xu. Multi-class token transformer for weakly supervised semantic segmentation. In CVPR, 2022.
>
> Jinheng Xie, Xianxu Hou, Kai Ye, and Linlin Shen. Clims: Cross language image matching for weakly supervised semantic segmentation. In CVPR, 2022.
>
> Qi Chen, Lingxiao Yang, Jian-Huang Lai, and Xiaohua Xie. Self-supervised image-specific prototype exploration for weakly supervised semantic segmentation. In CVPR, 2022.

---

> ### Author Response · Authors · 2022-11-18
> **Responses to Reviewer PPnr (3/4)**
>
> > Q3. *The experimental results are mixed or marginally improved compared to the WILSON framework. On some task settings of the VOC benchmark, the overall improvements are around 1\% and on some task settings of the COCO-to-VOC benchmark, the proposed method is inferior to the baseline.*
>
> **Answer**: We respectfully but strongly disagree that the performance of our proposed RaSP is only marginally improved over WILSON because the experimental results indicate otherwise. To dispel any misunderstanding, we would like to draw the reviewer's attention to the following tasks where we improve over WILSON. We restate the corresponding relative percentage gain in favour of our method. For instance, in Tab. 1 gains are (i) +1% in 15-5 VOC, (ii) +1.6% in 10-10 VOC, (iii) +4.0% in 10-5 VOC, (iv) +18.8% in 10-2 VOC, overlapped settings. A similar positive pattern can be observed in the Supplementary where in Tab. A2 for the VOC disjoint setting where gains are (i) +0.9% in 15-5, (ii) +0.7% in 10-10, (iii) +3.2% in 10-5, and (iv) +19.6% in 10-2. While in the setting of 60-20 COCO-to-VOC, our RaSP is outperformed by WILSON, our method shines when the number of incremental steps increases (e.g., +30.3% and +11.6% for COCO and VOC, respectively, in Tab. 2). Finally, in the hardest scenario of single-class incremental settings (see Tab. 3 and Tab. A3) RaSP consistently outperforms WILSON by significant margins. Similar trend can also be observed for several of the few-shot class-incremental scenarios in  Tab. 4 where RaSP outperforms WILSON.
>
> In summary, considering that our proposed RaSP surpasses WILSON numerous times, and many times by non-trivial margins, we beg to differ that RaSP yields mixed or marginally improved results compared to WILSON in the task of weakly supervised class-incremental segmentation.

---

> ### Author Response · Authors · 2022-11-18
> **Responses to Reviewer PPnr (2/4)**
>
> > Q2. *The assumption on the background class is problematic for incremental segmentation. This work seems to assume a static background class, which is not true in this task. Due to this background drifting issue, the proposed pseudo label generation strategy seems less effective: In equation (2), the bkg class would have a high similarity to previous bkg class which includes novel classes. This would largely suppress the new classes and conflict with the CAM loss. To make this work, it seems to require a good base segmentation network to generate foreground masks, which is difficult for the incremental setup.*
>
> **Answer**: Thank you for bringing up the issue about *background shift* in class-incremental segmentation. Indeed, we did not expand on this important discussion and would like to take this opportunity to elaborate on the background shift issue.
>
> From our experiments, we observe that the old model often confuses new class (*unseen*) objects with objects from the old classes, rather than with the background class (see Fig. 3 and Fig. A5). So besides the background shift issue, as per our observations, incremental segmentation also suffers from incorrect and overconfident predictions due to strong visual similarities among classes. We highlight this acute observation for the first time in the incremental segmentation task and root our method on this.
>
> To recap, we compute the semantic similarity maps (described in Eq. (2) of the main paper) only for the new foreground classes $\mathcal{C}^t$ present in an incremental step $t$. In other words, the semantic map $s^{bkg}$ for the *bkg* class is not computed, and not enforced by the optimization in Eq. (3). Moreover, we selectively backpropagate the RaSP loss $\mathcal{L}_{\text{RaSP}}$ only for those new class channels of the localizer $G_t$ for which ground truth *image labels* are available. As an example, in an incremental step $t$, if there are five new classes, $|\mathcal{C}^t| = 5$, and if for a given image only the new class ''dog'' is present, then we simply backpropagate the gradients of the RaSP loss for the ''dog'' channel only. All the other channels, including the *bkg* channel, are ignored during the backpropagation. Given the fact that the old model does not perfectly predict the new classes as *bkg* and is spuriously activated as foreground for the new classes (see the $(F \circ  E)^{t-1}({\bf x}_t)$ column in Fig. A5 where new class objects are not *bkg*), the RaSP loss in practice does not largely suppress the CAM loss. We have added a detailed discussion in the Sec. D of the Supplementary. We hope that our new findings will encourage future WSCI works to tackle overconfident model predictions on unseen classes.
>
> In principle, having a good base segmentation network is a plus in order to fully benefit from the proposed RaSP loss. However, in case of imperfect boundary predictions due to the absence of a strong starting base model, the pixel-level supervisory signal offered by the semantic similarity maps of our proposed method will still be stronger and denser with respect to the weakly supervised CAM loss  of WILSON (Eq. (4) of the main paper). Secondly, as explained in the [answer](https://openreview.net/forum?id=0vG8GbuPOH3&noteId=YiYykPNyXAH) to **Q1** of Reviewer **paY4**, we can reasonably assume that in practical applications dense annotations are available for commonly occurring primitive classes and we can expect that new classes will statistically have some resemblance to the older ones.

---

> ### Author Response · Authors · 2022-11-18
> **Responses to Reviewer PPnr (1/4)**
>
> We would like to thank Reviewer PPnr for acknowledging the timeliness of our work in addressing the less explored weakly supervised class-incremental segmentation (WSCI) task and the innovativeness of our work in promoting forward transfer.
>
> > Q1. *The overall novelty of the proposed framework is limited. It heavily relies on the existing framework WILSON and the main contribution is an improved WSSS loss integrated within each incremental step*
>
> **Answer**: Please refer to the [answer](https://openreview.net/forum?id=0vG8GbuPOH3&noteId=zmby_Kw0mJ-) to **Q2** from Reviewer **paY4**.

---

### Official Review · Reviewer_paY4 · 2022-10-27

**Confidence:** 4
**Clarity, Quality, Novelty And Reproducibility:** 1) The paper writing and organization…
**Correctness:** 3
**Technical Novelty And Significance:** 2
**Empirical Novelty And Significance:** Not applicable
**Recommendation:** 6

**Strength And Weaknesses:**

Strength:
1) The idea of using the semantic distance as a guidance is straightforward and has been verified effective.
2) The experimental evaluation is sufficient and reasonable.

Weakness:
1) The proposed method could not deal with the case that new class has rare similarity with existing classes. In addition, how to calculate the semantic distance is not clear and need more discussion.
2) The overall method is based on WILSON, which also weaken the contribution.

**Summary Of The Paper:**

This paper proposes a weakly supervised approach that leverages semantic relations to transfer the previously learned semantic class knowledge into the new class. The main motivation is the similar class often has closer distance in the feature space. Extensive experiments are conducted to validate this method.

**Summary Of The Review:**

Overall, this is a good paper with a novel class incremental learning that using semantic priors. Based on the strengths and the weakness, currently, the reviewer suggests a positive rating to this paper.

---

> ### Author Response · Authors · 2022-11-18
> **Responses to Reviewer paY4 (2/2)**
>
> > Q2. *The overall method is based on WILSON, which also weaken the contribution.*
>
> **Answer**: Our proposed semantic similarity RaSP loss is a general-purpose *plug-and-play* module and does not heavily rely on the WILSON (Cermelli *et al.*, 2022) framework. RaSP has been designed with the goal of improving forward transfer in incremental learning scenarios by converting *weaker* image label supervision into *denser* pixel-level supervision. As a matter of fact, our RaSP loss can be integrated into any weakly supervised semantic segmentation (WSSS) framework from the literature, as all we need is a neural network that outputs pixel-level predictions and the label names of the previously seen classes. We instantiate our contribution within WILSON because it is the state of the art in weakly supervised class-incremental segmentation (WSCI). In principle, RaSP can be seamlessly integrated with the WSSS baselines: vanilla CAM (Zhou *et.al.*, 2016b), SEAM (Wang *et al.*, 2020), SS (Araslanov \& Roth, 2020), and EPS (Lee *et al.*, 2021). It could be combined with any objective that mitigate *catastrophic forgetting*. In the experimental results we do not report those variants of RaSP and instead choose the existing state-of-the-art baseline of WILSON.
>
> We strongly believe that integrating our proposed RaSP with WILSON does not diminish its novelty. Rather, we view it as a simple and intuitive objective than can capitalize on the pathological predictions of any WSCI framework to its advantage, and improve the forward transfer in the weakly supervised incremental scenarios for free.
>
> **References**:
>
> Fabio Cermelli, Dario Fontanel, Antonio Tavera, Marco Ciccone, and Barbara Caputo. Incremental Learning in Semantic Segmentation from Image Labels. In CVPR, 2022.
>
> Bolei Zhou, Aditya Khosla, Agata Lapedriza, Aude Oliva, and Antonio Torralba. Learning deep features for discriminative localization. In CVPR, 2016b.
>
> Yude Wang, Jie Zhang, Meina Kan, Shiguang Shan, and Xilin Chen. Self-supervised equivariant attention mechanism for weakly supervised semantic segmentation. In CVPR, 2020.
>
> Nikita Araslanov and Stefan Roth. Single-stage semantic segmentation from image labels. In CVPR, 2020.
>
> Seungho Lee, Minhyun Lee, Jongwuk Lee, and Hyunjung Shim. Railroad is not a train: Saliency as pseudo-pixel supervision for weakly supervised semantic segmentation. In CVPR, 2021.

---

> ### Author Response · Authors · 2022-11-18
> **Responses to Reviewer paY4 (1/2)**
>
> We would like to thank Reviewer paY4 for appreciating the simplicity and the empirical effectiveness of our proposed method.
>
> > Q1. *The proposed method could not deal with the case that new class has rare similarity with existing classes. In addition, how to calculate the semantic distance is not clear and need more discussion.*
>
> **Answer**: It is indeed true that in edge-cases, where the new classes are very dissimilar to the old classes, the model can not benefit from the proposed RaSP loss. However, in practical applications we can reasonably assume that a model has already learned an array of *primitive* classes (often leveraging stronger pixel-level supervision), and that the incremental learner will encounter new objects that have some degree of resemblance to those primitive classes. Nevertheless, over a longer period of time, RaSP will eventually lead to efficient incremental learning with weaker supervision. Please refer to the [answer](https://openreview.net/forum?id=0vG8GbuPOH3&noteId=9ypAGpZgD2) to **Q3** of reviewer **AMCk** for an experiment with different class orderings, where we show that RaSP does not hurt performance in such scenarios.
>
> We agree that we should have included more details about the semantic similarity measure ${\bf S}_\Omega$ in our submission. The similarity metric used in our work is derived from the cosine distance, which is computed between a pair of class labels names as:
>
> \begin{align}
> {\bf S}_\Omega = - (1 - \frac{\omega(c_i) \cdot \omega(c_j)}{||\omega(c_i)||_2 ||\omega(c_j)||_2}).
> \end{align}
> where $\omega(c_i)$ and $\omega(c_j)$ represent the vectorial embeddings for the $i^\text{th}$ and $j^\text{th}$ classes. The value of ${\bf S}_\Omega$ is then substituted to Eq. (2) of the main paper. Note that the higher the semantic similarity between a pair of labels $c_i$ and $c_j$, the higher the $s^c_i$ value.
>
> We obtain the vectorial embedding $\omega(c)$ corresponding to a class label name $l_c$ using the BERT transformer (Devlin *et al.*, 2019). In details, we prompt a transformer with the class label name to obtain a 768-dimensional vector representation $\omega(c) = \texttt{Transformer}(''\texttt{An image of a }\{l_c\}'')$. While one could omit the prompt and simply provide the class label name, we do it to give context to the transformer that the class label name is a *noun*. Please note that our method can work with other semantic mapping functions, *e.g.*, Word2Vec (see Tab. A1 in the Supplementary). We have updated the Supplementary with these details; see Sec. A.1.
>
> **References**:
>
> Jacob Devlin, Ming-Wei Chang, Kenton Lee, and Kristina Toutanova. BERT: Pre-training of Deep Bidirectional Transformers for Language Understanding. In NAACL HLT, 2019.

---

### Author Response · Authors · 2022-11-18
**General Comments and Highlights from the Rebuttal**

We sincerely appreciate the effort from the reviewers for providing insightful comments, remarks and suggestions to improve the manuscript. In this rebuttal we would like to clarify the concerns raised by the reviewers. We have highlighted the changes in the Supplementary material made during the rebuttal period in 'brown'.

Before we delve into the point-by-point answer to the reviewers, we would like to highlight some key points from the rebuttal that are in response to the common points from the reviewers.

1. **Novelty**: We proposed a novel semantic prior loss, RaSP, that for the first time improves forward transfer in the relevant yet relatively less explored weakly supervised class-incremental segmentation (WSCI) task. Our loss exploits the old model predictions and the semantic relationships between the new and old classes to provide denser supervision to the new classes for *free*, which is otherwise not available in the WSCI task. RaSP is then integrated on top of an existing WSCI framework as a *plug-and-play* module, demonstrating its flexibility. Our contributions are not tied to this framework.

2. **Evaluation Settings**: Given the existing WSCI evaluation settings were limited, in this work we propose  an array of new incremental scenarios for rigorous analysis. In particular, we additionally evaluated on longer incremental tasks, fewer classes in each task (including learning one class per task), different class orderings, and few-shot incremental scenarios. The proposed RaSP and all competitors were evaluated using standard benchmarks established in the relevant research community.

3. **SOTA Performance**: Despite its simplicity, RaSP improves the existing state-of-the-art results in the WSCI task on multiple incremental scenarios, in several cases by non-trivial margins. We show that RaSP really shines in the harder incremental scenarios which involve learning over longer periods of time with fewer classes - the scenarios that one would commonly encounter in real world applications. Moreover, we showed that RaSP is also robust to different class orderings, and thus, potentially resilient to edge-cases that may occasionally arise in real applications.

---

### Decision · Program_Chairs · 2023-01-20

**Decision:**

Reject

**Justification For Why Not Higher Score:**

This is a well-written paper with a crisp narrative and convincing (in some aspects) empirical evaluation. However, the incremental nature of the contribution, combined with limitations of the experimental evaluation, make it fall short of the bar for acceptance in its current form.

**Justification For Why Not Lower Score:**

N/A

**Metareview: Summary, Strengths And Weaknesses:**

# Summary of Contribution

This paper describes a method for weakly-supervised class-incremental learning for continual semantic image segmentation (called RaSP). The authors propose to use image-level labels and pseudo-labels inferred using the previous model (and similarity of new classes and old measured in an embedding space) to perform class-incremental semantic segmentation. The proposed approach is integrated into the WILSON class-incremental semantic segmentation model for testing. Experimental results are given on PASCAL VOC and MS-COCO for a variety of class-incremental learning scenarios.

# Strengths

+ **Relevance**: The problem of weakly-supervised, incremental semantic image segmentation is an important problem that has relatively little coverage in the existing continual learning literature. Considering longer task sequences is also of relevant interest. The experimental evaluation demonstrates the effectiveness of the proposed integration of semantic similarity into WILSON for most considered scenarios.

+ **Clarity and Reproducibility**: The reviewers are nearly unanimous in their praise of the clarity of technical presentation and believe the results to be reproducible.

# Weaknesses

+ **Novelty**: The main contribution of RaSP is the introduction of semantic similarity maps between new and old classes, which is a relatively minor intervention to the localization prior in Eq. (4) of WILSON (Cermelli et al., 2022). The authors argued during the discussion phase that RaSP is a "plug-and-play" module, easily integrable in any weakly-supervised semantic segmentation model. To effectively claim this, additional experiments integrating RaSP with other state-of-the-art methods would be needed. As presented, it is a minor modification to WILSON.

+ **Marginal Improvements**: While integrating RaSP into WILSON indeed improves performance, most significantly on longer task sequences, these longer task sequences lack broader comparison with the state-of-the-art -- which has been evaluated mostly on 2-task sequences, for which the improvements of RaSP are less significant.

# Summary

This is a well-written paper that makes a contribution to incremental, weakly-supervised semantic image segmentation. However, the incremental nature of the main contribution is not enough to meet the bar for acceptance at ICLR. While the extension to consideration of longer task sequences is appreciated, considering WILSON and WILSON+RaSP in the comparative performance analysis is limits this as a contribution.